# Chronic Nutrition Impact Symptoms Are Associated with Decreased Functional Status, Quality of Life, and Diet Quality in a Pilot Study of Long-Term Post-Radiation Head and Neck Cancer Survivors

**DOI:** 10.3390/nu13082886

**Published:** 2021-08-22

**Authors:** Sylvia L. Crowder, Zonggui Li, Kalika P. Sarma, Anna E. Arthur

**Affiliations:** 1Department of Food Science and Human Nutrition, University of Illinois at Urbana-Champaign, 386 Bevier Hall 905 S Goodwin Ave, Urbana, IL 61801, USA; aarthur@illinois.edu; 2Department of Health Outcomes and Behavior, Moffitt Cancer Center, 4117 E Fowler Ave, Tampa, FL 33617, USA; 3Department of Psychology and Neuroscience, Boston College, 140 Commonwealth Ave, Chestnut Hill, MA 02467, USA; il@bc.edu; 4Carle Cancer Center, Carle Foundation Hospital, 602 W University Ave, Urbana, IL 61801, USA; Kalika.sarma@carle.com

**Keywords:** survivorship, head and neck, diet, symptoms, quality of life

## Abstract

Background: As a result of tumor location and treatment that is aggressive, head and neck cancer (HNC) survivors experience an array of symptoms impacting the ability and desire to eat termed nutrition impact symptoms (NISs). Despite increasing cancer survival time, the majority of research studies examining the impact of NISs have been based on clinical samples of HNC patients during the acute phase of treatment. NISs are often chronic and persist beyond the completion of treatment or may develop as late side effects. Therefore, our research team examined chronic NIS complications on HNC survivors’ functional status, quality of life, and diet quality. Methods: This was a cross-sectional study of 42 HNC survivors who were at least 6 months post-radiation. Self-reported data on demographics, NISs, quality of life, and usual diet over the past year were obtained. Objective measures of functional status included the short physical performance battery and InBody© 270 body composition testing. NISs were coded so a lower score indicated lower symptom burden, (range 4–17) and dichotomized as ≤10 vs. >10, the median in the dataset. Wilcoxon rank sum tests were performed between the dichotomized NIS summary score and continuous quality of life and functional status outcomes. Diet quality for HNC survivors was calculated using the Healthy Eating Index 2015 (HEI-2015). Wilcoxon rank sum tests examined the difference between the HNC HEI-2015 as compared to the National Health and Nutrition Examination Survey (NHANES) data calculated using the population ratio method. Results: A lower NIS score was statistically associated with higher posttreatment lean muscle mass (*p* = 0.002). A lower NIS score was associated with higher functional (*p* = 0.0006), physical (*p* = 0.0007), emotional (*p* = 0.007), and total (*p* < 0.0001) quality of life. Compared to NHANES controls, HNC survivors reported a significantly lower HEI-2015 diet quality score (*p* = 0.0001). Conclusions: Lower NIS burden was associated with higher lean muscle mass and functional, physical, emotional, and total quality of life in post-radiation HNC survivors. HNC survivors reported a significantly lower total HEI-2015 as compared to healthy NHANES controls, providing support for the hypothesis that chronic NIS burden impacts the desire and ability to eat. The effects of this pilot study were strong enough to be detected by straight forward statistical approaches and warrant a larger longitudinal study. For survivors most impacted by NIS burden, multidisciplinary post-radiation exercise and nutrition-based interventions to manage NISs and improve functional status, quality of life, and diet quality in this survivor population are needed.

## 1. Introduction

Head and neck cancer (HNC) is a heterogeneous disease including cancer of the oral cavity, oropharynx, hypopharynx, and larynx [1]. As a result of treatment and treatment that is aggressive, HNC survivors experience an array of symptoms impacting the desire and ability to eat termed nutrition impact symptoms (NISs) [2]. Common NISs include dysphagia, xerostomia, and difficulty chewing that lead to comprised food intake, malnutrition, and increased susceptibility to infection [2,3,4]. HNC survivors are living longer, thus increasing the survivorship period [2,5]. Despite increasing cancer survival time, the majority of research studies examining the impact of NISs have been based on clinical samples of HNC patients during the acute phase of treatment. NISs are often chronic and persist beyond the completion of treatment or may develop as late side effects [2]. Therefore, it is crucial for healthcare providers to examine chronic NIS complication on survivors’ functional status, quality of life, and diet quality.

While it has previously been established that the time around formal diagnosis until approximately three months post-treatment has been associated with decreased quality of life, little is known regarding the chronic burden, defined as greater than 6 months, of treatment-related outcomes in HNC survivors [6]. Quality of life is a measure of survivors’ overall well-being and often encompasses several domains—functional, physical, social, and emotional health. HNC is considered as one of the most emotionally traumatic cancer types [7] and impacts quality of life outcomes [8,9]. Studies have identified anxiety [10], depression [11], and psychological problems [12] in HNC survivors to be associated with poorer quality of life. However, few studies have examined the chronic impact of aggregated NIS burden on overall and domain-specific quality of life and functional status. This is of the upmost importance as one of the most important quality of life factors is nutrition [13] and adaptations to eating and psychological concerns regarding dysphagia and xerostomia may further reduce functional status in survivors [2,14,15]. The HNC population faces unique nutritional challenges as compared to other cancer types, likely decreasing functional status and quality of life.

Before interventions can be designed to enhance survivorship outcomes of long-term survivors, data is needed to inform researchers that provides evidence of reduced quality of life, functional status, and diet quality in long-term HNC survivors that identify critical targets for interventions. Therefore, the objective of this pilot study was to evaluate the relationship between an NIS summary score on quality of life and functional status and compare the diet quality of post-radiation head and neck cancer (HNC) survivors to age-matched National Health and Nutrition Examination Survey (NHANES) controls.

## 2. Subjects and Methods

### 2.1. Design

This was a cross-sectional study of 42 HNC survivors who were previously diagnosed or treated in a Midwestern hospital within six months to nine years post treatment. In the quality of life model, the dependent variables of interest were total, emotional, physical, functional, and social quality of life. In the functional status model, the dependent variables of interest were body mass index, functional status composite score, body fat percentage, and lean muscle mass. The independent variable of interest was aggregated NIS burden reflected by a dichotomized NIS summary score. Dietary intake for HNC survivors was assessed using the Healthy Eating Index-2015 (HEI-2015) [16] collected with the semi-quantitative Harvard food frequency questionnaires (FFQ) [17] as compared to NHANES controls using the population-ratio method [18]. All study activities were approved by the Institutional Review Boards of Carle Foundation Hospital and the University of Illinois at Urbana-Champaign (Project ID: 17088; UIUC number: 181933; Original Approval Date: 1 February 2018) and adhered to the principles of the Declaration of Helsinki. All participants were informed of the purpose and procedures of the study and informed written consent was obtained from all participants before data collection.

### 2.2. Study Population

Participant screening and recruitment occurred between March 2018 and May 2019. Criteria for eligibility included: (1) Previous diagnosis of stage I–IV primary cancer of the oropharynx, hypopharynx, larynx, or oral cavity; (2) within 6 months to 10 years posttreatment with radiation; (3) no evidence of disease, deemed by oncologist and/or surgeon; (4) ability to consume food orally; (5) ≥18 years of age; and (6) English-speaking. HNC survivors treated at the hospital were identified via the Hospital Cancer Registry and a letter was mailed to potential participants explaining the research study. Participants were called within 2–3 weeks of receiving the mailed letter. Medical records were searched to prevent calling deceased individuals. Interested participants were then scheduled for an in-person study visit. At the study visit, formal written consent was obtained.

### 2.3. Procedures

Participants completed a self-administered health survey that included data on demographics, behavioral characteristics, quality of life, and NIS burden. Dietary data were obtained using the self-administered 2007 Harvard FFQ [17,19,20]. Functional status data were obtained using the short physical performance battery [21] and InBody© body composition testing [22]. An electronic medical record (EMR) review was conducted to collect clinical data on cancer stage, treatment type, and time since diagnosis. Participants were compensated $50.

## 3. Measures

### 3.1. Predictor: Nutrition Impact Symptoms

The Functional Assessment of Cancer Therapy-Head and Neck (FACT-H&N) Additional Concerns (AC) Subscale was used to measure perceived NIS barriers [23]. The scale consists of 12 questions, six specifically referring to NIS including: (1) ability to eat any foods desired, (2) ability to eat as much as desired, (3) no presence of xerostomia, (4) ability to swallow naturally and easily, (5) ability to eat solid foods, and (6) no presence of pain in the mouth, throat, or neck, with answers ranked on a 5-point Likert scale ranging from 0 (not at all) to 4 (very much). The scale was coded so that a lower score indicated fewer disease- or treatment-related symptoms. The individual symptom scores were summed to create a total overall NIS summary score (range 4–17) and dichotomized as ≤10 vs. >10, the median in the dataset.

### 3.2. Outcome Variable: Functional Status

A study team member trained in anthropometrics collected the following functional status measures: anthropometrics, bioelectrical impedance analysis (BIA) [22], and the short physical performance battery [21].

Anthropometric measures for height were conducted in accordance with the Anthropometric Standardization Reference Manual [24]. Height was collected by a trained research staff member during the study visit and measured to the nearest 0.5 inch (without shoes).

Measures of body composition were determined by a vertical direct segmental multi-frequency BIA analyzer (InBody© 270, Cerritos, CA, USA). The InBody© 270 records a user’s weight, lean muscle mass, body mass index, and percent body fat to the nearest 0.1 lb (without shoes and in light clothing with pockets emptied). The method of measuring body composition via a BIA has been previously validated and used in similar clinical studies [22]. Four participants were unable to complete InBody© measures as a result of other health conditions (pacemakers and physical impairments).

The short physical performance battery consists of three functional tests assessing performance. The physical performance battery has been previously used in the HNC population [21]. Tests of standing balance include tandem, semi-tandem, and side-by-side stands. Walking speed tests included an 8-foot walking course, with no obstructions for an additional two feet at either end. Participants were allowed to use assistive devices when needed, and each participant was timed for two walks [25]. To test the ability to rise from a chair (termed chair rise-and-sits), a straight-backed chair was placed next to a wall; participants were asked to fold their arms across their chest and to stand up and sit down five times as quickly as possible [25]. Participants were timed from the initial sitting position to the final sitting position at the end of the fifth stand. The short physical performance battery provides a summed composite score based on the number of seconds able to hold a semi-tandem, tandem, and/or side-by-side stance with feet together; 8-foot walk time; and time to complete 5 chair rise-and-sits [25].

### 3.3. Outcome Variable: Quality of Life

Overall quality of life was assessed using the FACT-H&N quality of life questionnaire (Cronbach’s coefficient alpha = 0.86) [23]. The FACT-H&N assesses the impact of cancer diagnosis and therapy in four subdomains: physical, social, emotional, and functional. The FACT-H&N has 28 questions, with answers ranked on a 5-point Likert scale ranging from 0 (not at all) to 4 (very much). The scale was coded so that a higher score indicated higher quality of life. Questions were summed to create four subdomain scores and the four subdomains were summed to create a continuous total summary score.

### 3.4. Outcome Variable: Dietary Intake

HNC survivors’ dietary intake information was collected using the validated 131 item semi-quantitative Harvard Food Frequency Questionnaire, which includes standard portions sizes for each item and the frequency of consumption over the past year [26]. The Harvard FFQ is a feasible instrument suited to assess associations of usual dietary intake and has been extensively used as a measure of dietary exposure in cancer studies [27]. The FFQ allows participants to choose the average frequency of consumption of food items over the past year from a Likert scale. Healthy eating of the HNC survivors was assessed using the HEI-2015 dietary measurement and compared to the NHANES controls using the population ratio method [16,28]. This method was employed because advice from the Dietary Guidelines for Americans (DGA) is designed to be met over time and this method best encompasses that intent for diet evaluation [29]. The HEI was developed to assess diet quality issued by the United States Department of Agriculture (USDA) based on the standards of a healthy lifestyle in association with health outcomes [30]. HEI is composed of 13 scored components and include 5 major food groups: fruit (total and whole), vegetable (total and greens/beans), grains (total and whole), dairy or alternative dairy and protein, oils, and nuts; in addition to limiting saturated fats, sodium, and empty calories [16]. Nine of the components focus on adequacy (dietary components to increase) and four focus on moderation (dietary components to decrease, including refined grains, sodium, added sugars, and saturated fats) [16]. The daily intakes for each component were standardized for energy by diving each study participant’s daily component intake by his or her total daily energy intake in kilocalories and multiplying by 100 prior to applying the HEI-2015 scoring algorithm [16]. Each of the 13 components of the HEI-2015 had a minimum score of zero and a maximum score ranging from 5 to 10 that reflected a pre-established level of intake [16]. The total HEI score is the sum of the components, with a range of 0 to 100 [16]. A score between 0 and 50 indicates a poor diet; 51 and 80, a moderate diet quality that needs improvement; and a score greater than 80, a good diet [16].

## 4. Statistical Considerations and Analyses

Descriptive statistics (means and frequencies) were generated for demographic and clinical variables. ANOVA tests were computed to detect statistical difference between quality of life and demographic variables. The FACT-H&N scoring manual was used to calculate the mean score and standard deviations [31]. Summary scores and subscale scores were extracted into three tertiles of the actual range of scores and categorized as mild, moderate, or severe using methods described in similar studies [32,33]. For example, in the FACT-G, the actual range is 0 to 108; therefore, the three tertiles were 0–36 as severe impairment, 37–72 as moderate impairment, and 73–108 as mild impairment [33].

For the functional status model, Wilcoxon rank sum tests between lean muscle mass, body fat percentage, body mass index, and functional status and the dichotomized NIS summary score were computed. For the quality of life model, Wilcoxon rank sum tests between subdomain and total quality of life measures and the dichotomized NIS summary score were computed. Wilcoxon rank sum tests examined the difference among HNC HEI-2015 as compared to NHANES controls using the population ratio method. Statistical significance was set as an alpha level <0.05. Statistical analyses were performed using SAS software, version 9.4 or later [34].

## 5. Results

### 5.1. Participant Characteristics

The overall demographic characteristics of the study population are shown in Table 1. The mean age of the study population was 62 years old, and most participants were married (57%). Most participants were white males (59.5%), with at least some college education (62%). The most common tumor location was the oral cavity, and most participants were diagnosed with stage III-IV cancer (30%). The majority of participants were 1–4 years post treatment (64%). The majority were former smokers (57%) and current alcohol users (48%). Accrual was met within 14 months and required screening of 266 HNC cases. Of these, 79 were eligible for study participation and 187 were ineligible or excluded. Of the 79 eligible HNC cases, 42 agreed to participate for an enrollment rate of 52.2%. Reasons for ineligibility were distance (*N* = 15), timing (*N* = 13), too sick (*N* = 6), and too busy (*N* = 3).

### 5.2. Functional Status and Nutrition Impact Symptoms

Table 2 reports the associations between functional status measures and the NIS summary score. A lower NIS summary score was statistically associated with higher post-treatment lean muscle mass (*p* = 0.002). A higher NIS summary score was non-statistically associated with a higher post-treatment body mass index and functional status composite score.

### 5.3. Quality of Life and Nutrition Impact Symptoms

Table 2 also reports associations between quality of life measures and the NIS summary score. A lower NIS summary score was statistically associated with higher functional (*p* = 0.0006), physical (*p* = 0.0007), emotional (*p* = 0.007), and total (<0.0001) quality of life.

### 5.4. Mean Quality of Life Summary Score

All FACT-H&N mean summary scores and FACT-H&N subscale scores were in the mild category (higher score), except for the Nutrition Impact Symptom Subscale (NIS) and Head and Neck Specific Concerns Subscale (HNCS) 6 item and 10 item, in which participants scored in the moderate category (Table 3).

### 5.5. Healthy Eating Index 2015 (HEI-2015) HNC Population vs. NHANES Data

As compared to NHANES controls, HNC survivors reported a significantly lower total HEI-2015 diet quality score (*p* = 0.0001) (Table 4). In the adequacy component, HNC survivors reported statistically significant lower consumption of total vegetables (*p* < 0.0001), whole grains (*p* < 0.01), total protein foods (*p* < 0.0001), seafood and plant proteins (*p* < 0.0001), and fatty acids (*p* < 0.0001). HNC survivors reported statistically significant higher consumption of dairy products (*p* < 0.01) and non-statistically significant higher consumption of total fruits and whole fruits. In the moderation component, HNC survivors consumed significantly more refined grains (*p* < 0.0001) and sodium (*p* < 0.0001), and significantly less added sugars (*p* < 0.0001) and saturated fats (*p* < 0.01) as compared to NHANES data.

## 6. Discussion

To our knowledge, this study was among one of the first to explore the chronic complications of self-reported NIS burden on quality of life, objective measures of functional status, and diet quality in HNC survivors greater than 6 months post treatment. Notable findings were that higher post-treatment quality of life scores were associated with a lower NIS summary score (lower NIS burden). Furthermore, higher post-treatment lean muscle mass was associated with a lower NIS summary score, suggesting those who reported lower symptom burden had higher functional capacity. As compared to NHANES controls using the HEI-2015 population ratio method, HNC survivors in our study consumed a statistically significant lower total overall diet quality, which may be a result of NIS burden impacting the ability and desire to eat, though further longitudinal studies exploring this association are warranted.

Associations between quality of life measures and NIS were explored. Findings indicated that a lower NIS summary score was significantly associated with higher physical, functional, emotional, and total quality of life. Surprisingly, NIS burden was not associated with social quality of life for long-term survivors. A study by List et al. examining pre-treatment coping strategies, reported that social support-seeking was the most common coping strategy used by patients with HNC and commonly begins immediately following diagnosis [35]. Therefore, it may be possible that quality of life outcomes, such as social well-being, improve in the months and years after treatment as it is likely long-term HNC survivors have had sufficient time to adapt to their new normal and seek social support groups [36].

Previous research has suggested self-reported impairments in functional performance are common in HNC survivors [37]. Despite their high prevalence, few multidisciplinary rehabilitation programs designed for HNC exist in the peer-reviewed literature, and in those studies, survivors were assessed during and immediately after treatment [38,39]. Our study was among one of the first to objectively measure functional status in post-treatment HNC survivors. Despite our small sample size, higher post-treatment lean muscle mass was associated with a lower NIS summary score. Furthermore, while our findings were not statistically significant, higher body mass index and functional status were non-statistically associated with a lower NIS summary score. Power calculations suggest a sample size of *N* = 80 HNC survivors would detect a significant difference between the groups [40]. This work encourages larger, more robust clinical trials assessing risk factors for symptom development coupled with exercise and nutrition-based rehabilitation in long-term survivors to improve functional capacity.

Given the emphasis on the totality of the diet by national guidelines, our study team examined diet quality using the HEI-2015 among HNC cancer survivors as compared to NHANES data. HNC survivors reported a significantly lower total diet quality score as compared to NHANES controls, a possible consequence of treatment-related NIS burden. HNC survivors had higher diet quality scores among the adequacy component—dairy. As HNC survivors often experience difficulties with certain textures and flavors, we hypothesize this increase is likely due to HNC survivors’ preference for soft foods, such as yogurt, which can be easily blended into smoothies and supplement drinks [41]. Additionally, the HNC population consumed nearly double the NHANES population for sodium and far less added sugars, likely a consequence of taste dysfunction resulting in food preference and aversion that is commonly reported in qualitative literature [2,4,42,43].

Limitations of this study should be noted. The cross-sectional design of the project is a limitation as quality of life and functional status measures were taken at only one time point. A prospective cohort study would be able to determine changes in different survivorship phases. Additionally, there is likely respondent bias inherent in the individuals in the individuals willing to complete the research study. The study population was selected from one Midwestern cancer center; consequently, the participants may not be fully representative of the total population of HNC survivors. The subjective bias of the FACT-H&N is a limitation of the study. Because the direction of the relationship between symptom burden and quality of life is unclear, objective measures are needed to determine if symptoms, such as swallowing dysfunction, result in quality of life declines or if declining quality of life emphasizes perceived symptom burden. Additionally, while the NISs examined were from a validated quality of life questionnaire, the NIS summary score was created for this specific analysis and is not validated.

This pilot study consisted of 42 HNC survivors. Although this study was severely underpowered, significant findings using simple Wilcoxon rank sum tests were noted. Furthermore, survivorship time ranged from 6 months to 10 years post treatment. Therefore, there was great heterogenicity in the population. A larger, longitudinal, and sufficiently powered study using multivariable regression models could further confirm directionality of quality of life, functional status, and diet quality changes post treatment based on this study’s preliminary findings. Other strengths of the study include the use of validated questionnaires and objective measures of functional status in addition to the a priori approach to characterizing diet quality.

## 7. Conclusions

Self-reported NIS impairments were associated with lower quality of life and functional status outcomes among a population of long-term HNC survivors. As compared to an age-matched population from NHANES, HNC survivors reported lower overall diet quality, likely a result of symptoms impacting the ability and desire to eat. Multidisciplinary post-radiation exercise and nutrition-based interventions to manage NISs and improve quality of life, functional status, and dietary intake in this vulnerable survivor population are needed.

## Figures and Tables

**Table 1 nutrients-13-02886-t001:** Demographic and clinical characteristics *N* = 42.

Characteristic	Total Participants
Age: Mean ± SD [range], years	62.7 ± 11.8 (32–81)
Body Mass Index: Mean ± SD [range], kg/m^2^	26.2 ± 4.85 (16.5–38.3)
Under/normal weight *N* (%)	21 (50.0)
Overweight/obese *N* (%)	21 (50.0)
Lean muscle mass ^a^: *N* (%)	
Under/normal	27 (71.0)
High	11 (29.0)
Body fat percentage ^a^: *N* (%)	
Under/normal	13 (34.2)
High	25 (65.8)
Gender: *N* (%)	
Male	25 (59.5)
Female	17 (40.5)
Ethnicity: *N* (%)	
Non-Hispanic	42 (100)
Race: *N* (%)	
European American/White	39 (92.9)
Other	3 (7.1)
Education: *N* (%)	
High school or less	16 (38.1)
Some college or more	26 (61.9)
Annual household income: (dollars/year) *N* (%)	
Less than $54,999	24 (57.1)
$55,000 or more	18 (42.9)
Marital Status: *N* (%)	
Married	24 (57.1)
Not married	18 (42.9)
Smoking Status: *N* (%)	
Current	3 (7.1)
Former	24 (57.2)
Never	15 (35.7)
Alcohol Status: *N* (%)	
Current	20 (47.7)
Former	19 (45.2)
Never	3 (7.1)
Time since diagnosis: *N* (%)	
<1 to 4 years	27 (64.3)
≤4 to 9 years	15 (35.7)
Tumor site: *N* (%)	
Oral cavity	20 (47.6)
Pharynx/Larynx	22 (52.4)
Cancer stage: *N* (%)	
I–II	12 (28.6)
III–IV	30 (71.4)
Treatment: *N* (%)	
Concurrent chemoradiation	26 (61.9)
Radiation only	16 (38.1)
Nutrition Impact Symptom Score	
NIS ≤10	15 (35.7)
NIS > 10	27 (64.3)

^a^*N* = 38.

**Table 2 nutrients-13-02886-t002:** Nutrition impact symptom burden and associated quality of life and functional status outcomes in head and neck cancer survivors *N* = 42.

Quality of Life Outcome	Mean (SD)	Median	*p*-Value ^a^
Functional QOL			0.0006 ^b^
NIS < 10	24.4 (3.7)	16.0	
NIS > 10	17.8 (6.9)	26.0	
Physical QOL			0.0007 ^b^
NIS < 10	25.1 (2.6)	26.0	
NIS > 10	20.5 (5.8)	21.0	
Emotional QOL			0.007 ^b^
NIS < 10	21.2 (2.9)	22.0	
NIS > 10	18.5 (3.8)	19.0	
Social QOL			0.09
NIS < 10	22.4 (5.7)	23.5	
NIS > 10	20.0 (7.2)	20.5	
Total QOL			0.0001 ^b^
NIS < 10	93.0 (11.7)	95.5	
NIS > 10	76.8 (14.2)	73.5	
**Functional Status Outcome**	**Mean (SD)**	**Median**	***p*-Value ^a^**
Lean muscle mass ^c^			0.002 ^b^
NIS < 10	75.7 (17.1)	76.4	
NIS > 10	59.9 (15.0)	55.8	
Body fat percentage^c^			0.26
NIS < 10	26.4 (9.0)	25.3	
NIS > 10	28.8 (9.9)	25.5	
Body mass index			0.18
NIS < 10	27.0 (5.1)	25.3	
NIS > 10	25.2 (4.5)	24.9	
Functional Status Score			0.18
NIS < 10	9.7 (2.5)	10.0	
NIS > 10	9.4 (1.9)	9.0	

^a^ Wilcoxon rank sum test; ^b^ Indicates statistical significance; ^c^
*N* = 38.

**Table 3 nutrients-13-02886-t003:** Functional Assessment Cancer Therapy (FACT) summary and subscale scores *N* = 42.

	# Items	Actual Range	Observed Range	Mean (SD)	Impairment Category
FACT summary scores					
FACT-General	27	0–108	52–107	85.31 (15.21)	Mild
FACT-Head and neck	37	0–148	65–135	105.19 (19.59)	Mild
FACT subscale scores					
Physical well-being	7	0–28	5–28	22.90 (4.91)	Mild
Social well-being	7	0–28	0–28	21.26 (6.47)	Mild
Emotional well-being	6	0–24	10–24	19.90 (3.58)	Mild
Functional well-being	7	0–28	7–28	21.24 (6.34)	Mild
Nutrition impact symptom questions	6	0–24	3–22	12.93 (5.26)	Moderate
Head and neck specific concerns	10	0–40	7–32	19.88 (6.39)	Moderate

**Table 4 nutrients-13-02886-t004:** Healthy Eating Index-2015 (HEI-2015) Carle HNC survivors vs. NHANES data.

Component	Actual Max Points	Carle HNC Survivors *N* = 42	Age-Matched NHANES Population 2015–2016 Data ^a^
Total HEI Score	100	54.3	60.6 ^b^
Adequacy			
Total Fruits	5	2.9	2.8
Whole Fruits	5	3.2	4.4
Total Vegetables	5	2.1	3.8 ^b^
Greens and Beans	5	3.4	3.7
Whole Grains	10	2.5	3.0 ^c^
Dairy	10	6.9	5.4 ^b^
Total Protein Foods	5	3.0	5.0 ^b^
Seafood and Plant Proteins	5	3.0	5.0 ^b^
Fatty Acids	10	3.3	4.5 ^b^
Moderation			
Refined Grains	10	10.0	7.2 ^b^
Sodium	10	8.9	3.7 ^b^
Added Sugars	10	0.5	6.9 ^b^
Saturated Fats	10	4.4	5.3 ^c^

^a^ Wilcoxon Rank Sum test to test difference; ^b^
*p* < 0.0001; ^c^
*p* < 0.01.

## Data Availability

The dataset generated and analyzed in the current study are not publicly available due to the sensitive nature of responses, but deidentified data are available from the corresponding author on reasonable request.

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
