# Peer review of "Chronic Nutrition Impact Symptoms Are Associated with Decreased Functional Status, Quality of Life, and Diet Quality in a Pilot Study of Long-Term Post-Radiation Head and Neck Cancer Survivors"

_nutrients, 2021, doi:10.3390/nu13082886_

Round 1

Reviewer 1 Report

The topics are interesting,  but in my opinion,  authors should  write In results :

  1. other analyses between groups depending ON :

             - sex (male and female),

             - level of education,

             - annual  income,

             - marital status,

             - time since diagnosis,  

                - tumor site,

                - cancer stage,

              - cancer treatment

  1. in table. 1 “[%]” should l be in another place
  2. the used terms must be clearer  

      Line 217: Most participants were white males, with at least some college    education (how many patients)

Lines 217 and 218: most participants were diagnosed with stage IV cancer (how many).

Lines 218 and 219: The majority of participants were 1-4 years post-treatment.  The majority were former smokers and currently alcohol users (how many and to what degree).

  1. In tab. 3 lack of number of survivors who had NIS<10 or NIS>10, authors write only the total number

Reviewer 2 Report

Thank you for the opportunity to review the manuscript, “Chronic nutrition impact symptoms are associated with decreased functional status, quality of life and diet quality in a pilot study of long term, post-radiation head and neck cancer survivors,” submitted to Nutrients.  The authors provide an informative cross-sectional study of long-term/chronic head and neck cancer survivors at a Midwestern US cancer center.  The authors utilize a nutritional impact scale to account for the feeding problems encountered in head and neck cancer survivorship, and, in what this reviewer would consider a reasonable sample size, demonstrate that impaired long-term/chronic (6m to 9 years post cancer treatment) impairment in feeding is associated with long term dietary abnormalities, body composition of changes, and quality of life changes.  These changes are generally in the direction that would be expected with impaired feeding.  The study design and analysis appear to be appropriate, and the authors appear to have demonstrated that long-term impairment of feeding is present after head and neck cancer treatment (expected).  They further seem to have adequately demonstrated that increasingly impaired patients experience nutritional sequela, body composition of sequelae and quality of life sequelae.  The impact of this study is that it highlights the importance of monitoring and treating head and neck cancer patients long-term (throughout their survivorship) to mitigate important measures of clinical deterioration as measured by body composition, dietary composition, and quality of life.
